# RedCaps: Web-curated image-text data created *by the people, for the people*

**Karan Desai**      **Gaurav Kaul**      **Zubin Aysola**      **Justin Johnson**

University of Michigan

{kdexd,kaulg,aysola,justincj}@umich.edu

https://redcaps.xyz

## Abstract

Large datasets of paired images and text have become increasingly popular for learning generic representations for vision and vision-and-language tasks. Such datasets have been built by querying search engines or collecting HTML alt-text – since web data is noisy, they require complex filtering pipelines to maintain quality. We explore alternate data sources to collect high quality data with minimal filtering. We introduce RedCaps – a large-scale dataset of 12M image-text pairs collected from Reddit. Images and captions from Reddit depict and describe a wide variety of objects and scenes. We collect data from a manually curated set of subreddits, which give coarse image labels and allow us to steer the dataset composition without labeling individual instances. We show that captioning models trained on RedCaps produce rich and varied captions preferred by humans, and learn visual representations that transfer to many downstream tasks.

## 1   Introduction

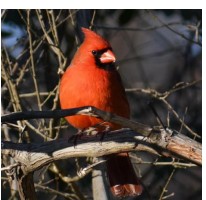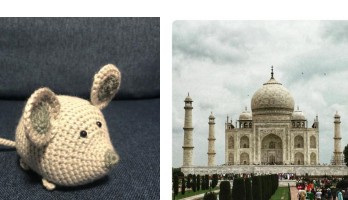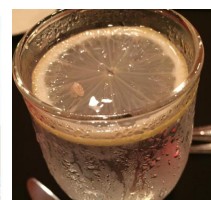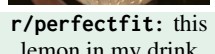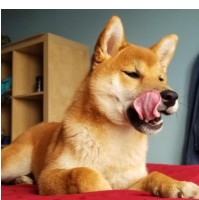

**r/birdpics:** male northern cardinal      **r/crafts:** my mom tied this mouse      **r/itookapicture:** itap of the taj mahal      **r/perfectfit:** this lemon in my drink      **r/shiba:** mlem!

Figure 1: **RedCaps dataset** comprises 12M image-text pairs from 350 subreddits. RedCaps data is created *by the people, for the people* – it contains everyday things that users like to share on social media, for example hobbies (**r/crafts**) and pets (**r/shiba**). Captions often contain specific and fine-grained descriptions (northern cardinal, taj mahal). Subreddit names provide relevant image labels (**r/shiba**) even when captions may not (mlem!), and sometimes may group many visually unrelated images through a common semantic meaning (**r/perfectfit**).

Large datasets of image-text pairs from the web have enabled successful transfer learning applications in computer vision. Two such prominent datasets – SBU [1] and Conceptual Captions [2] – are widely used for pre-training vision-and-language (V&L) representations [3–11] that transfer to a variety of downstream V&L tasks like visual question answering [12–14], visual reasoning [15, 16], and image captioning [17, 18]. Recent work [19, 20] also shows that image-text data from COCO [17] can be used to learn *visual* features that are competitive with supervised pretraining [21] on ImageNet [22, 23] when transfered to downstream tasks [24–28]. More recently, CLIP [29] and ALIGN [30] scale up to 400M and 1B+ web-curated image-text pairs, enabling zero-shot visual recognition.

35th Conference on Neural Information Processing Systems (NeurIPS 2021) Track on Datasets and Benchmarks.

These datasets have an appealing advantage – they are free from expensive annotations. However, they apply complex filtering steps to deal with noisy web data. For example, Conceptual Captions (CC-3M [2], CC-12M [31]) discard captions without nouns, or whose nouns do not match with image labels predicted by in-house image taggers. They also perform text pre-processing like replacing proper nouns with common nouns. These pipelines are data-inefficient – for example, CC-3M collected 5B image-text pairs and filtered them down to 3.3M. CLIP and ALIGN scale primarily by *relaxing* such filtering, resulting in gargantuan datasets which could be extremely noisy.

How can we obtain high-quality image-text data from the web *without* complex data filtering? We argue that the quality of data depends on its *source* and the *intent* behind its creation. Revisiting data sources, SBU query Flickr with predefined keywords while CC-3M and CC-12M extract images and HTML alt-text from an unspecified set of web pages; CLIP and ALIGN give only vague descriptions of their data sources, and their datasets are non-public. In these sources, text is secondary to images: Flickr focuses on photos, and alt-text is an oft-overlooked *fallback* when images cannot be viewed that frequently contains metadata or generic text (e.g. "alt img" [30]). To obtain higher-quality data, we look for sources where humans use both images and text equally for interaction on the web.

In this paper, we explore the Reddit [32] social media platform for collecting image-text pairs. Textual data from Reddit is already used for pre-training massive language models [33–36] in NLP. We collect images and their captions as submitted by Reddit users in topic-specific subreddits. Our dataset of image captions from Reddit (RedCaps in short) consists of 12M image-text pairs submitted in 350 subreddits between 2008–2020. RedCaps data is created *by the people, for the people* to engage with the broader community. Figure 1 shows some examples from RedCaps – the captions are more conversational, humorous, emotional, and generally more diverse than HTML alt-text.

Apart from linguistic diversity, Reddit offers many other advantages. Subreddits provide additional image labels and group related content – manually selecting subreddits allows us to steer dataset contents without labeling individual instances. Reddit's *voting* system gives free and organic quality control: unappealing or spam content is actively *downvoted* by users or removed by moderators. RedCaps is one of the largest public image-text datasets, but it is not *static*: we plan to release regular updates with newly uploaded Reddit content, allowing RedCaps to *grow* over time.

We claim that captions written with the intent of human interaction on Reddit are a better source of data than used in other image-text datasets. To this end, we follow VirTex [19] to learn visual representations by training image captioning models from scratch. We find that human evaluators prefer captioning outputs from models trained on RedCaps vs CC-3M. We also transfer the learned features to **eleven** different downstream datasets for tasks including image classification, object detection, instance segmentation, and fine-grained recognition using both fine-tuning and language-based zero-shot classification [29]. We show that features learned on RedCaps outperform those learned on SBU or CC-3M, demonstrating the utility of our data collection strategy.

## 2 RedCaps: Collecting image-text pairs from Reddit

Reddit is the singular data source for RedCaps. This leads to a very different data collection pipeline than datasets based on HTML alt-text or search engine results. Here we describe how we collect RedCaps.

**Overview of Reddit:** Reddit is a social media platform for content sharing and discussion. It comprises user-run communities called *subreddits* that cover diverse topics like animals (r/cats, r/foxes), food (r/pizza, r/sushi), leisure (r/hiking, r/crafts), and utility (r/ceramics, r/tools). Users can submit new posts or share existing posts from other subreddits (*cross-posting*), and may comment and upvote (or downvote) posts to express their interest.

We are specifically interested in posts containing images. Figure 2 shows an image post submitted by user u/johndoe in subreddit r/itookapicture. It com-

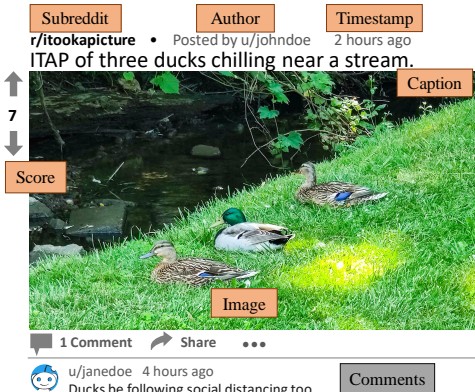

Figure 2: **Preview of a Reddit image post:** We build RedCaps by extracting images and metadata (in orange) from such image posts.

prises an image, caption, score (upvotes minus downvotes), and information about the author and time of post creation. We extract this metadata from millions of image posts to build RedCaps.

Reddit posts also have associated comment threads. These are usually casual conversations *loosely* based on the image. In Figure 2, the comment describes ducks as following *social distancing* – it includes context beyond the image (COVID-19 pandemic) and conveys it with a witty remark. Prior works in dialog modeling and text summarization have trained on Reddit comments [33, 37–40]. For RedCaps, we only use captions as textual data and leave comments for future work.

## 2.1 Data collection pipeline

Reddit's uniform structure allows us to parallelize data collection as independent tasks – each task involves collecting posts submitted to a single subreddit in one year. Our collection pipeline has three steps: (1) subreddit selection, (2) image post filtering, and (3) caption cleaning.

**Step 1. Subreddit selection:** We collect data from a manually curated set of subreddits. Subreddits have their own rules, community norms, and moderators so curating subreddits allows us to steer the dataset's composition without annotating individual instances. We select subreddits with a high volume of images posts, where images tend to be photographs (rather than memes, drawings, screenshots, etc) and post titles tend to describe image content (rather than making jokes, political commentary, etc). We do not select any NSFW, banned, or quarantined subreddits. We want to minimize the number of *people* that appear in RedCaps, so we omit subreddits whose primary purpose is to share or comment on images of people (such as celebrity pics or user selfies). We choose subreddits focused on general photography (r/pics, r/itookapicture), animals (r/axolotls, r/birdsofprey, r/dachshund), plants (r/roses, r/succulents), objects (r/classiccars, r/trains, r/mechanicalkeyboards), food (r/steak, r/macarons), scenery (r/cityporn[1], r/desertporn), or activities (r/carpentry, r/kayaking). In total we collect data from 350 subreddits; the full list can be found in Appendix A.

**Step 2. Image post filtering:** We use Pushshift [41] and Reddit [42, 43] APIs to download all image posts submitted to our selected subreddits from 2008–2020. Posts are collected at least six months after their creation to let upvotes stabilize. We only collect posts with images hosted on three domains: Reddit (i.redd.it), Imgur (i.imgur.com), and Flickr (staticflickr.com). Some image posts contain multiple images (*gallery posts*) – in this case we only collect the first image and associate it with the caption. We discard posts with $< 2$ upvotes to avoid unappealing content, and we discard posts marked NSFW (by their authors or subreddit moderators) to avoid pornographic or disturbing content.

**Step 3. Caption cleaning:** We expect Reddit post titles to be less noisy than other large-scale sources of image captions such as alt-text [2, 31], so we apply minimal text cleaning. We lowercase captions and use ftfy [44] to remove character accents, emojis, and non-latin characters, following [29, 35, 36]. Then we apply simple pattern matching to discard all sub-strings enclosed in brackets ((.*), [.*]). These sub-strings usually give non-semantic information: *original content* tags [oc], image resolutions (800x600 px), camera specs (shot with iPhone), self-promotion [Instagram: @user], and other references (link in comments). Finally, like [31] we replace social media handles (words starting with '@') with a [USR] token to protect user privacy and reduce redundancy. Due to such filtering, ≈12K (0.1%) captions in our dataset are empty strings. We do not discard them, as subreddit names alone provide meaningful supervision. Unlike CC-3M or CC-12M that discard captions without nouns or that don't overlap image tags, we do not discard any instances in this step.

Through this pipeline, we collect 13.4M instances from 350 subreddits. Our collection pipeline is less resource-intensive than existing datasets – we do not require webpage crawlers, search engines, or large databases of indexed webpages. RedCaps is easily extensible in the future by selecting more subreddits and collecting posts from future years. Next, we perform additional filtering to mitigate user privacy risks and harmful stereotypes in RedCaps, resulting in final size of 12M instances.

## 2.2 Ethical considerations

There has been growing awareness about potential biases and harms that can arise from internet-scale image and text datasets [45–51]. There is a fundamental tension in such datasets: the use of internet data is motivated by the desire to use datasets larger than can be manually annotated or verified, but this also means that such datasets cannot be fully controlled or curated by their creators.

---

[1]Many subreddits are jokingly titled *-porn* to indicate beautiful non-pornographic images.

We identify two potential risks with RedCaps – privacy of people appearing in RedCaps images, and harmful stereotypes – and attempt to minimize them by *automatic data filtering*. We also discuss the impact of data curation from Reddit on user consent and data distribution in RedCaps.

**Privacy:** The individual who *posts* a given photo on Reddit may not be the person *appearing* in said photo; this can pose privacy risks for people who did not expect to appear in images online [49, 50]. Our first method of mitigation is the manual curation of subreddits which are not focused on describing people (Section 2.1). As an additional measure, we use RetinaFace [52] to filter images having any face detection with confidence $\geq 0.9$. Results of this filtering are shown in Table 1. The number of detections are high (1.2M), however the precision is low (32%) – most detections are masked faces, statues, and animals. Nevertheless we remove all of these images to reduce privacy risks while minimizing impact to downstream vision tasks.

| | Detected | Precision | | Missed dets. | |
|---|---|---|---|---|---|
| | (Filtered) | 5K | (%) | 50K | 12M |
| Faces | 1.2M | 1615 | 32% | 79 | ≈19K |
| NSFW | 87K | 65 | 1% | 1 | ≈240 |
| Language † | 24K | – | – | – | – |

Table 1: **Automatic filtering:** We use detectors to filter ∼1.4M instances with images containing faces or NSFW content, or captions containing potentially derogatory language. We estimate the *precision* of these detectors by reviewing 5K random detected images. After filtering, we review 50K random images (out of 12M) to estimate *missed detections* – faces and NSFW images remaining in RedCaps – which we find to be extremely low.
†*: Language filtering is deterministic (string matching).*

**Harmful Stereotypes:** Another concern with Reddit data is that images or language may represent harmful stereotypes about gender, race, or other characteristics of people [48, 49, 51]. We select only non-NSFW subreddits with active moderation for collecting data. This stands in contrast to less curated uses of Reddit data, such as GPT-2 [35] whose training data includes at least 63K documents from banned or quarantined subreddits which may contain toxic language [53]. We attempt to further reduce harmful stereotypes in two ways:

- **NSFW images:** We use the InceptionV3 [54] model from [55] to filter images detected as *porn* or *hentai* with confidence $\geq 0.9$. Similar to face filtering, we estimated precision of our filtering and estimated amount of missed detections, shown in Table 1. The model detects 87K images with low precision (∼1%) – most detections are non-NSFW images with pink and beige hues.
- **Potentially derogatory language:** We filter instances whose captions contain words or phrases from a common blocklist [56]. It is important to note that such coarse filtering might suppress language from marginalized groups reclaiming slurs [51]; however, as RedCaps is not intended to describe people, we believe this is a pragmatic tradeoff to avoid propagating harmful labels.

**Consent:** When submitting to Reddit, users expect their posts to be publicly visible and accessible via the Reddit API we use to download data. However, they did not explicitly consent for their data to be used for training large-scale neural networks [49]. We mitigate this concern in two ways. First, we distribute URLs instead of images; posts deleted from Reddit will thus be automatically removed from RedCaps. Second, we provide a public form allowing anyone to request that specific instances be removed from RedCaps on our website. These decisions mean that over time some image will disappear from RedCaps, making it difficult to *exactly* reproduce experiments in the future. However we believe this to be less important than allowing users to opt out from RedCaps. Even if images are removed, we expect RedCaps to *grow* over time as we include newer posts (Figure 3).

**Reddit demographics:** Reddit's user demographics are not representative of the population at large. Compared to US adults, Reddit users skew male (69% vs 49%), young (58% 18-29 years old vs 22%), college educated (36% vs 28%), and politically liberal (41% vs 25%) [57]. Reddit users are predominantly white (63%) [57], and 49% of desktop traffic to Reddit comes from the United States [58]. All of the subreddits in RedCaps use English as their primary language. Taken together, these demographic biases likely also bias the types of objects and places that appear in images on Reddit, and the language used to describe these images. We do not offer explicit countermeasures to these biases, but users of RedCaps should keep in mind that *size doesn't guarantee diversity* [51].

Subtler issues may also exist, such as imbalanced representation of demographic groups [59] or gender bias in object co-occurrence [60] or language [61]. These are hard to control in internet data, so we release RedCaps with explicit instructions on suitable use-cases; specifically requesting models not be trained to identify people, or make decisions that impact people. We document these instructions and other terms-of-use in a datasheet [45], provided in Appendix G.

# 3 RedCaps data analysis

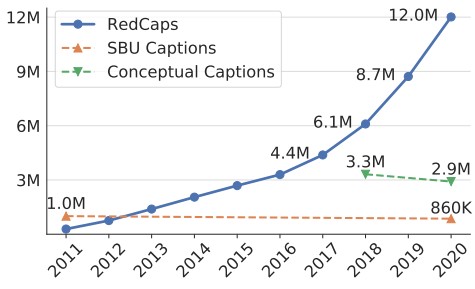

| Datasets in 2021 | # Instances | Released |
|---|---|---|
| RedCaps (ours) | 12,011,111 | ✓ |
| CC-12M [31] | 12,423,374 | ✓ |
| WIT-english [62] | 5,500,746 | ✓ |
| CLIP [29] | 400M | ✗ |
| ALIGN [30] | 1.8B | ✗ |

Figure 3: **Dataset size comparison:** RedCaps is one of the largest public image-text datasets, and is expected to *grow* over time.

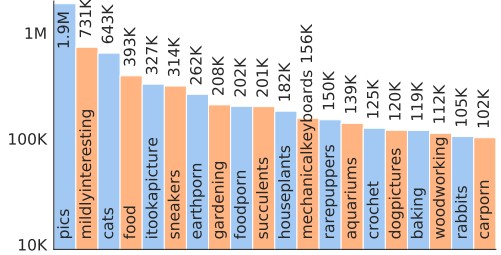

Figure 4: **Instances per subreddit:** Top 20 subreddits with most image-text pairs in RedCaps.

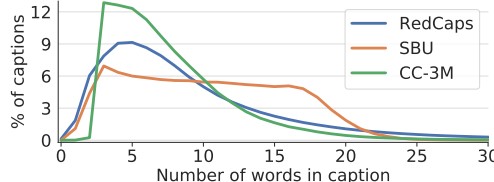

Figure 5: **Caption Lengths:** RedCaps has a long tailed distribution of caption lengths.

**Dataset size:** Figure 3 (top) shows the growth of RedCaps between 2011–2020 based on creation timestamps of image posts (see Figure 2). We observe that both SBU and CC-3M have shrunk in size since their release. Since these datasets have released images as URLs (similar to us), an instance would become invalid if the underlying image is removed from the URL [2]. Likewise, some instances in RedCaps can also disappear in the future if Reddit users delete their posts. However, new image posts on Reddit outnumber deleted posts – we expect RedCaps size to increase in future versions.

Figure 3 (bottom), compares RedCaps with recent image-text datasets released in 2021. RedCaps is 2× larger than the English subset of multilingual Wikipedia image-text dataset [62], and nearly as large as CC-12M [31]. Based on current trends, we expect RedCaps to outsize CC-12M by the end of 2021. While CLIP [29] and ALIGN [30] used orders of magnitude larger training datasets, they are not released for public use – RedCaps remains one of the largest public image-text datasets.

**Subreddit distribution:** RedCaps instances are distributed across 350 subreddits in a long-tail distribution. In Figure 4, we show top 20 subreddits with most instances in RedCaps. Subreddit sizes highly correlate with their popularity on Reddit, which depends on what users find interesting to view and share on social media. Large subreddits are based on general photography (r/pics, r/mildlyinteresting, r/itookapicture), while specific subreddits show that Reddit users enjoy sharing images of food (r/food, r/foodporn), cute pets (r/cats, r/dogpictures, r/rabbits), and show off their hobbies (r/gardening, r/crochet, r/baking) and accesories (r/sneakers, r/mechanicalkeyboards, r/carporn). This gives a distribution of visual concepts encountered by humans in daily life without having to predefine an ontology of object classes.

**Caption lengths:** Figure 5 compares caption lengths between RedCaps and other datasets. We see that RedCaps has the highest mode length at 5 words (vs 3 for CC-3M, SBU) and a heavier tail of long captions ≥25 words. SBU has a fairly flat distribution of captions between 3 and 17 words, likely since they only retain captions with at least one preposition and two words in a manually curated term list; RedCaps and CC-3M captions are not filtered in this way and have more peaked distributions reflecting natural language usage.

**Word count statistics:** Table 2 (top) compares linguistic diversity between datasets by computing the number of unique unigrams (words), bigrams, and trigrams occurring at least 10 times. This reveals that CC-3M has surprisingly little linguistic diversity, having less unique unigrams than SBU despite having ≈3× *more* captions. RedCaps has the most unique terms, with more than 4× unigrams and more than 3× bigrams and trigrams than CC-3M. Greater linguistic diversity means that models trained on RedCaps should recognize a larger variety of visual concepts.

---

[2]We use full SBU and CC-3M annotations for analysis instead of discarding captions with invalid URLs.

| Dataset | Unigrams | Bigrams | Trigrams |
|---------|----------|---------|----------|
| SBU | 28,989 | 107,847 | 99,687 |
| CC-3M | 21,223 | 230,077 | 287,017 |
| RedCaps | 95,777 | 770,100 | 866,243 |

| | Top-5 frequent Trigrams |
|---------|-------------------------|
| SBU | in front of, black and white, in the sky in the background, in the water |
| CC-3M | a white background, on a white, image may contain, illustration of a may contain person |
| RedCaps | itap of a, i don't, one of my itap of the, this is my |

Table 2: **Word count statistics:** Number of $\{1, 2, 3\}$-grams occurring at least 10 times **(top)** and top-5 trigrams in each dataset **(bottom)**.

| Dataset | C. Nouns | P. Nouns | Adjectives | Verbs |
|---------|----------|----------|------------|-------|
| SBU | 12,985 | 8,748 | 2,929 | 2,497 |
| CC-3M | 8,116 | 654 | 4,676 | 3,467 |
| RedCaps | 26,060 | 38,405 | 11,029 | 6,019 |

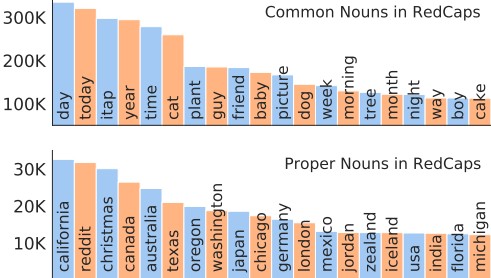

Figure 6: **Linguistic statistics:** Number of unique words by POS, occurring at least 10 times **(top)**, and frequent nouns in RedCaps **(bottom)**.

Table 2 (bottom) shows the most frequent trigrams per dataset. SBU has many prepositional phrases, likely since they require all captions to contain a preposition. Common CC-3M trigrams *image may contain*, *may contain person* suggest that the alt-text from which CC-3M takes captions may sometimes be automatically generated. RedCaps trigrams *I don't*, *one of my*, *this is my* are more conversational and draw a personal connection between the author and the image, whereas other trigrams *itap of a* and *itap of the* reflect community conventions on r/itookapicture.

**Linguistic statistics:** We use part-of-speech (POS) tagging to dig deeper into linguistic diversity of RedCaps. We use the en_core_web_trf model from SpaCy [63] to tag POS in all captions. Figure 6 (top) shows number of unique words per POS appearing at least 10 times. RedCaps has $>2\times$ more common nouns and $>4\times$ more proper nouns than SBU, and $>2\times$ more adjectives and $>1.5\times$ more verbs than CC-3M. Nouns in CC-3M are artifically deflated, since their pipeline replaces proper nouns and named entities with hypernyms (which may explain their low unigram counts in Table 2).

Figure 6 (bottom) shows the most frequent occurring nouns in RedCaps. We see a variety of common nouns, both concrete (*cat*, *plant*) and abstract (*day*, *time*). We find that nouns like *guy*, *baby*, and *boy* are frequent with RedCaps images with pet animals. Moreover, most frequent proper nouns comprise many cities (*chicago*, *london*), states (*california*, *texas*), and countries (*japan*, *germany*, *india*), indicating the geographical diversity of RedCaps.

## 4 Experiments

We aim to show that RedCaps offers a unique style of data for both vision and V&L applications. We demonstrate both applications by adapting VirTex [19], a recent method for pre-training visual representations by performing image captioning as proxy task. In this section, we measure the effect of data quality on downstream vision tasks by training VirTex models with the same architecture but different datasets – SBU, CC-3M, and RedCaps. To control for RedCaps's size, we also train on a subset of RedCaps instances from 2020 – this has size comparable to CC-3M (*3.2M vs 2.9M*).

**Extending VirTex to VirTex-v2:** VirTex comprises an image encoder (*visual backbone*) and a pair of text decoders (*textual head*) that predict the caption token-by-token in forward and backward directions. The base model from [19] used a ResNet-50 [21] visual backbone, and Transformers [64] in textual head that are $L = 1$ layers deep and $H = 2048$ dimensions wide, and was trained on COCO Captions [17] (118K images). We modify this model from [19] to VirTex-v2 in order to scale to larger noisy datasets, making the following changes:

– **Model architecture:** We use deeper Transformers with $L = 6$ layers. To balance the memory requirements, we reduce the width to $H = 512$. We use the recent *Pre-LN* Transformer variant [35, 65, 66] that is more stable to train large transformers [67] – LayerNorm [68] is moved inside the residual connection, and we add LayerNorm before the prediction layer.

| Pre-train Dataset | Pets | Food | Flowers | Cars | Country | SUN | Birdsnap | **Average Accuracy** |
|---|---|---|---|---|---|---|---|---|
| | *N = 37* | *N = 101* | *N = 102* | *N = 196* | *N = 211* | *N = 397* | *N = 500* | |
| **Zero Shot** SBU | 8.7 | 3.0 | 13.7 | 0.6 | 0.6 | 14.7 | 1.3 | 6.1 |
| CC-3M | 15.5 | 10.9 | 10.1 | 0.5 | 0.5 | **33.3** | 1.6 | 10.3 |
| RedCaps-20 | 41.8 | **54.6** | **33.5** | **3.2** | 2.3 | 23.9 | **11.8** | **24.4** |
| RedCaps | **42.4** | 53.8 | 26.2 | 3.1 | **3.6** | 26.8 | 8.3 | 23.5 |
| **Lin. Probe** SBU | 61.8 | 48.5 | 80.3 | 22.2 | 12.0 | 61.3 | 18.6 | 43.5 |
| CC-3M | 69.9 | 57.3 | 76.6 | 25.2 | 12.8 | **70.0** | 16.1 | 46.8 |
| RedCaps-20 | **87.0** | 79.1 | 85.9 | 39.1 | 11.6 | 63.6 | **30.6** | 56.7 |
| RedCaps | 85.0 | **80.8** | **86.3** | **43.9** | **13.6** | 67.3 | 28.1 | **57.9** |

Table 3: **Transfer learning: zero-shot and linear probe.** We train VirTex-v2 models on different image-text datasets, then transfer the learned features to seven downstream classification datasets (*N = #classes*). Models trained on RedCaps perform best on all datasets except one.

– **Tokenization:** Similar to VirTex, we use SentencePiece tokenizer [69] with BPE [70]. We build a vocabulary of 30K tokens from the combined caption corpus of SBU, CC-3M and RedCaps. For fair comparison, we use the same vocabulary for all models trained on different datasets. When training with RedCaps, we *prefix* the caption with subreddit tokens: e.g. for Figure 1 (`r/birdpics`), the caption becomes `[SOS] bird pics [SEP] northern male cardinal [EOS]`. We use `wordsegment` [71] to break subreddit names to words (e.g. itookapicture → i took a picture).
– **Training details:** We use AdamW [72, 73] with weight decay $10^{-2}$ and max learning rate $5 \times 10^{-4}$ with linear warmup for the first 10K iterations, followed by cosine decay [74] to zero. We also use label smoothing ($\epsilon_{ls} = 0.1$) [54] which has improved language generation for machine translation [64]. We train for 1.5M iterations with total batch size 256 across 8× 2080Ti GPUs.

We save checkpoints every 2000 iterations, and average the last five checkpoints to use for downstream tasks and image captioning. All other details remain unchanged from [19]. We have open-sourced all the training code and pre-trained checkpoints, available at `https://redcaps.xyz`.

## 4.1 Transfer learning on downstream vision tasks

We evaluate the quality of visual representations learned from SBU, CC-3M, and RedCaps by training VirTex-v2 models on each, then transferring the visual backbone to image classification and instance segmentation on **eleven** different downstream datasets. Our evaluation setup closely follows recent works on self-supervised learning [75–77] and language-supervised [19, 29] learning. We describe the main evaluation settings here; see Appendix F for more details.

**Zero-shot image classification:** Training with language supervision enables *zero-shot* transfer to downstream tasks without *any* task-specific training [29, 78]. We evaluate the utility of different datasets for representation learning by comparing zero-shot performance on seven classification datasets: Oxford-IIIT Pets [79], Food-101 [80], Flowers-102 [81], Stanford Cars [82], Country-211 [29], and SUN-397 [83], and Birdsnap [84]. Inspired by CLIP [29], we perform zero-shot classification by designing one *prompt* per category in the target dataset and ranking the log-probabilities predicted by the trained captioning model for each prompt, averaging predictions from the forward and backward Transformers. For SBU and CC-3M we follow CLIP and use the prompt `[SOS] a photo of a/an _ [EOS]`; for RedCaps we adjust to the training setup and use a prompt with prefixed subreddit – `[SOS] i took a picture [SEP] itap of a/an _ [EOS]`.

Results are shown in Table 3 (top). VirTex-v2 models trained on RedCaps outperform those trained on SBU and CC-3M by *wide* margins on **six** out of seven datasets. This not due to RedCaps's larger size: models trained on RedCaps-20 also outperform those trained on CC-3M.

**Linear probe image classification:** We also evaluate image classification on these datasets by training linear models over *frozen* visual features. Our evaluation details exactly follow CLIP – we use `scikit-learn` [85] logistic regression with L-BFGS. We train for 1K iterations, and search L2 regularization $\lambda$ over 96 logarithmic spaced values in $[10^{-6}, 10^6]$ by validating on held-out 10% training data. Results are shown in Table 3 (bottom) with similar trends as zero-shot transfer.

**Comparison with CLIP:** Despite improvements over SBU and CC-3M, our absolute zero-shot performance falls behind CLIP (e.g Food-101 top-1 with ResNet-50 – 81.1 vs. 54.6). Their results are not comparable, as CLIP uses a different architecture (contrastive vs autoregressive), deeper

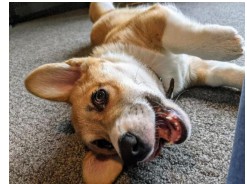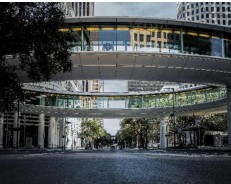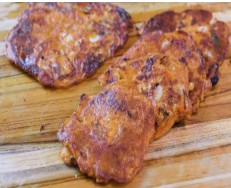

| | | | | |
|---|---|---|---|---|
| CC-3M | animal lying on the ground | a car is completely covered in snow. | the building is a-story polished concrete floor. | how to cook a rack of ribs |
| RedCaps | **r/lookatmydog:** my little guy | **r/mildlyinteresting:** this snow sculpture | **r/pics:** a building in singapore | **r/foodporn:** homemade pizza |

Figure 7: **Human evaluation: CC-3M vs. RedCaps.** We decode image captions from VirTex-v2 models trained on CC-3M and RedCaps. We show both captions (excluding subreddit names) to three crowd workers and ask them to guess which is more likely to be written by a human. All three workers chose the underlined caption for each of the displayed images. We found that workers preferred organic references (little guy vs animal), witty remarks (snow sculpture), and specific mentions (singapore) by the RedCaps-trained model. Among negative cases are mostly instances where RedCaps-trained models make blatant errors in identifying common visual objects (e.g. pizza).

transformer (12 vs 6 layers), larger dataset (400M vs 12M instances), longer training (12.8B image updates vs 384M), and prompt ensembling. Our goal is not to achieve state-of-the-art performance, but instead to compare impact of different data sources on the quality of learned visual features.

**Other tasks:** We evaluate on standard transfer tasks with four other datasets: PASCAL VOC and ImageNet-1k linear classification with *frozen* features and instance segmentation [86] on COCO [26] and LVIS [27] with *end-to-end fine-tuning* of Mask R-CNN. These tasks follow the same setup as [19]. On ImageNet, we also perform $k$ nearest neighbor classification ($k=20$), following [87, 88], and zero-shot classification as described above. Results are shown in Table 4. All models

| Pre-train Dataset | ImageNet Top-1 | | | VOC | COCO | LVIS |
|---|---|---|---|---|---|---|
| | Zero shot | Linear Cls. | k-NN (k=20) | Cls. mAP | Segm. AP | Segm. AP |
| SBU | 5.2 | 45.5 | 38.7 | 85.0 | 36.5 | 22.0 |
| CC-3M | 20.7 | **53.9** | 45.4 | 87.0 | **37.2** | 22.9 |
| RedCaps | **22.7** | 53.4 | **52.0** | **87.5** | 37.0 | **23.0** |

Table 4: **Additional tasks:** RedCaps trained model matches or exceeds models trained on SBU/CC-3M.

perform similarly on fine-tuning tasks (COCO and LVIS), while RedCaps trained model gains on tasks involving minimal or no fine-tuning – k-NN (52.0 vs 45.4) and zero-shot (22.7 vs 20.7) on ImageNet, and linear classification on VOC (87.5 vs 87.0).

## 4.2 Image captioning

We hope that the human interaction flavored data of RedCaps enables more human-like and *conversational* image captioning models. We use VirTex-v2 pre-trained models for image captioning – we use nucleus sampling [89] with nucleus size 0.9 to decode a caption from the forward Transformer. In this section, we demonstrate all results on an additional *held-out test set* of 1K instances sampled randomly from image posts submitted to our selected subreddits in the first week of 2021.

**Evaluating caption predictions:** Automatic captioning evaluation metrics correlate poorly with human judgement [90, 91]. We thus evelute caption predictions via user studies. We sample captions from models trained on RedCaps and CC-3M, then present crowd workers with the image and both captions. Workers are told that one caption is written by a human and the

| RedCaps vs. | RedCaps preferred |
|---|---|
| CC-3M | 63.3% |
| Human | 41.6% |

other machine-generated, and asked to guess which is human-written. We take a majority vote among three workers for each of our 1K test images. Results are shown to the right – workers preferred captions from the RedCaps-trained model for 633/1000 images. We run a similar study to compare against ground-truth captions, and workers still prefer generated captions for 416/1000 images. Some qualitative results are shown in Figure 7; more are shown in Appendix (Figure 10).

**Subreddit-conditioned generation:** Captions from different subreddits have distinct styles, focusing on different image aspects or using community-specific jargon. We use this observation to generate captions with distinct styles by prompting a RedCaps-trained model with *different* subreddits. Figure 8 shows examples of such diverse captions for images; see Appendix (Figure 11) for more.

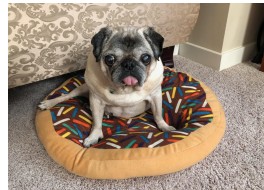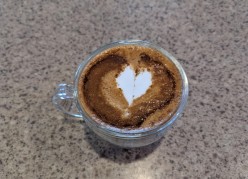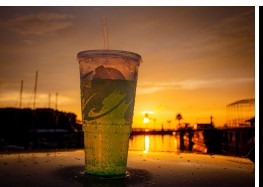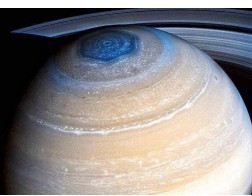

| **r/itookapicture:** itap of my dog. | **r/itookapicture:** itap of my coffee | **r/earthporn:** sunset in venice, italy | **r/earthporn:** saturn's north pole. |
| **r/absoluteunits:** this absolute unit of a pug | **r/absoluteunits:** this absolute unit of a coffee. | **r/food:** a cold beer on the beach. | **r/food:** the clearest image of saturn |
| **r/somethingimade:** i made a bed for my pug. | **r/somethingimade:** i made a heart latte. | **r/pics:** shot from a beach house! | **r/pics:** the clearest image of saturn ever taken |

Figure 8: **Subreddit-controlled caption style.** We prompt the VirTex-v2 model trained on RedCaps with subreddit names while decoding captions. We observe that such conditioning captures subtle linguistic structures (**r/itookapicture**: itap of ..., **r/somethingimade**: i made...). or changes the main subject of caption (**r/earthporn**: venice, **r/food**: cold beer). However, for completely unrelated images (saturn), the model tends to ignore the conditioning while generating captions.

## 5 Related work

RedCaps is directly related to many recent efforts on building large datasets of image-text pairs from the internet without expensive human annotation. Two notable datasets are SBU [1] and Conceptual Captions [2]. Originally intended for image-text retrieval and image captioning, they are now widely used for training generic V&L representations [3–11, 92] that transfer to downstream tasks like visual question answering [12–14], referring expressions [93], and visual reasoning [15, 16]. More recent works build larger datasets specifically for V&L pre-training, e.g. LAIT [94], Conceptual-12M [31], and Wikipedia-ImageText [62]. Similar to these datasets, RedCaps offers rich semantic data for pre-training applications. However, our choice of data source and hence the data quality is unique.

Image-text datasets are now also used for learning visual features. Li et al. [78] trained visual N-gram models on YFCC-100M [95]; [19, 20] learn features from COCO Captions [17] that are competitive with supervised ImageNet training [21, 96] on many downstream tasks [22, 24, 26–28], and [29, 30] scale up to very larger non-public datasets that are larger than RedCaps.

A core motivation for collecting image-text data is scaling to larger datasets without bearing annotation costs. Related to this goal are efforts that learn from large quantities of noisy non-text labels for web images such as WebVision [97], YFCC-100M [95], JFT-300M [98, 99], and Instagram-3.5B [100].

## 6 Conclusion

This paper has introduced RedCaps, a large-scale dataset of images and captions collected from Reddit. As a source of data, Reddit is appealing: text and image are both created and shared by people, for the explicit purpose of starting a discussion with other people, leading to natural and varied content. Its subreddit structure allows manually curation of our dataset's content without labeling individual instances. We utilize this structure to collect a dataset focused on animals, objects, scenery, and activities, and specifically aim to minimize the appearance of people. We have shown that RedCaps is useful for learning visual representations that transfer to many downstream tasks, including zero-shot settings that use no task-specific training data. We have also shown that RedCaps can be used to learn image captioning models that generate high-quality text of multiple styles.

RedCaps is not without flaws. We have tried to minimize problematic content through subreddit curation and automated filtering, but the unfathomable nature of large data means that RedCaps may contain a small number of instances with NSFW images or harmful language. Reddit's demographic biases mean that RedCaps may not equally represent all groups. Users should carefully consider these limitations for any new tasks developed on RedCaps, and should be especially wary of applications that make predictions about people. Despite these limitations, we hope that RedCaps will help enable a wide variety of new applications and advances in vision and language.

## Acknowledgments

We thank Mohit Virli for suggestions on the project website. We thank Mohamed El Banani, Nilesh Kulkarni, Stefan Lee, Ramprasaath Selvaraju, Ramakrishna Vedantam, and Erik Wijmans for helpful discussions and feedback on the paper. We thank Priya Goyal and Ishan Misra for help related to VISSL codebase. We thank all anonymous reviewers for constructive feedback during the review phase. We also thank the UMich ARC-TS team for support with GPU cluster management.

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
