# OpenReview forum: "RedCaps: Web-curated image-text data created by the people, for the people"
_NeurIPS.cc/2021/Track/Datasets_and_Benchmarks/Round1 — NeurIPS 2021 Datasets and Benchmarks Track (Round 1)_

### Official Review · Reviewer_ugYV · 2021-06-30
**Promising contribution with major ethical concerns**

**Rating:** 5
**Confidence:** 4

**Strengths:**

1. **[Major] Language diversity & curation**. I find the author's claim that the captions and images are more "conversational, humorous, emotional, and generally more diverse" than typical HTML alt-text - I agree that the authors have identified a uniquely rich source for images and image captions compared to prior datasets. Linguistic statistics in 3.1 back this claim.
2. **[Minor] Human evaluation**. The authors conduct a helpful study to compare the readability/attractiveness of a captioning model trained on their dataset and find an improvement over previous models. The use of Reddit scoring is another unique component of the dataset that many developers & researchers might find useful.
3. **[Minor] Documentation and disclaimers**. The authors cite and follow relevant data ethics literature (especially Prabhu & Birhane) in presenting the risks of their data to potential data users (though I discuss later how those concerns should be better addressed).

**Weaknesses:**

1. **[Major] As yet unaddressed ethical concerns.** The authors correctly and clearly identify many ethical concerns and propose several solutions (mostly automated filtering) that they *plan* to implement before public release at the end of 2021, but have only forecasted the results of those mitigation techniques. The paper lacks a thorough test of the efficacy of those solutions with respect to both dataset composition (does the dataset still contain a large number of NSFW, stereotypical, offensive or otherwise problematic images?) and model performance (do models trained on this dataset exhibit problematic behavior?). The Ethics section of this review raises further concerns about the sufficiency of the proposed techniques and the ways the authors choose to balance ethical trade-offs.
2. **[Minor] Usefulness of stylistic captions for other tasks.** While this dataset has the distinct advantage of relying on stylized, conversational captioning, this also poses a weakness that the authors don't clearly state: captions produced by a model trained on this dataset may be less useful for less playful image captioning applications, or applications where phrases like "itap" are out of context - though reading the paper, there is some indication that the dataset is rich enough to enable a model to discern between contexts given enough subreddits (e.g. Fig. 8). Asking the human evaluators to mark which caption they prefer doesn't capture which caption is more useful or informative - this is a limitation. An increase in stylistic speech also make things more difficult for non-English speakers.

EDIT (07/16/2021): the authors have helped clarify concern (2) somewhat. See the ethics section for an update on (1).

**Additional Feedback:**

The dataset website looks very nice! Well done.

**Clarity:**

The paper is clearly written and easy to follow. The same applies to the documentation. The figures are useful, well annotated, and clearly relevant.

**Correctness:**

As far as I can tell, the collection and sanitization methods presented are correct and reproducible. The collection method is laudably simple and easy to follow. The authors take care to include failsafe measures in case of bad data - for example, the authors use both moderator-created NSFW ratings and NSFW ratings from their own automated classification model.

**Documentation:**

There is sufficient detail on collection, organization, and availability. Notes about maintenance are somewhat vague - there are many cases when the authors promise "we *will*" to implement a certain safeguard (such as the opt out form) before public release even though they haven't already done so. I would expect all of these promises to be fulfilled before publication.

The authors are also vague about the intended uses of their dataset, from an ethical standpoint: in the datasheet, they state that the dataset shouldn't be used for production systems that will make decisions that impact people. If this is really the case, it would seem the dataset isn't very useful at all - and the authors explicitly test an image captioning model with human annotators!

EDIT (07/16/2021): The authors helped clarify this in a statement that I expect will be included in the datasheet/documentation.

**Ethics:**

**Model training without ethical precautions.** In lines 205-210, the authors acknowledge that due to "computational constraints," all the models demonstrated in the paper were trained on the unfiltered version of the dataset containing high levels of NSFW images, faces, and derogatory language. Most concerning, these models were trained before the data creators (Reddit users) were given the option to opt-out, and their results were presented to other humans (crowd workers). Were the samples presented to crowd workers checked for stereotypes? Ethical precautions should be taken before data use, unless it is for explicit impact assessment or other ethics-related testing. It's not clear to me what computational constraints would prevent this.

**Lack of bias testing.** There is lots of literature on bias testing for image datasets, including Prabhu & Birhane (2021) - cited by the authors - such as methods for [analyzing co-occurrence statistics]([https://arxiv.org/abs/2004.07999](https://arxiv.org/abs/2004.07999)). The authors apply an NSFW classifier but fail to explore other aspects of dataset fairness. One technique to consider would be inclusive people annotations to allow more comprehensive fairness analysis, as in [Schumann et al. (2021)]([https://arxiv.org/pdf/2105.02317.pdf](https://arxiv.org/pdf/2105.02317.pdf)).

For example, I scrolled the "beachporn" category of the dataset for ~2 minutes and found several depictions of women dressed in beachwear. As a result, a model trained on this dataset may learn to disproportionately sexualize women, as in [this study]([https://dl.acm.org/doi/10.1145/3442188.3445932](https://dl.acm.org/doi/10.1145/3442188.3445932)) of ImageNet. It's not clear whether this might occur without further study of this and other stereotypes - and it's not clear that the proposal to filter NSFW content would cover this kind of subtle stereotype. These kinds of associations, if they exist, should be identified and disclaimed.

**Deferred responsibility.** While the authors acknowledge many bias issues and attempt to address them, issues of privacy & fairness are deferred to the data user in too many cases. Because the authors distribute URLs and not images (which has the benefit of allowing Reddit users to remove their data by deleting their posts), they can't ensure that faces are blurred. I don't think it's acceptable to defer this responsibility to developers.

**Opt-out, not opt-in**. The authors chose an opt-out strategy (users can delete their posts or fill out form), but as far as I can tell don't provide any notice to users that their images are being used for this dataset. Without notice, how likely is the opt-out option to be exercised? Ethically, I would strongly prefer an explicit opt-in notification mechanism, though I acknowledge that many recent privacy laws do not require this.

**Unacceptable levels of risk.** While I would prefer more stereotype testing, I am impressed by the authors' thorough investigation of NSFW content and identifiable faces. However, I'd like a more thorough discussion of the implications and acceptable levels of risk for this data.

The authors estimate that in the best case (after filtering and in the case that developers bother to blur or remove all detected faces), there still remain 61,000 adult faces and 2,000 child faces in the dataset. Prima facie, I am inclined to think there should be *zero* unblurred faces, especially given the proposed opt-out mechanism. Likewise, there are still 1,000 NSFW images that could be offensive or stereotypical (see Prabhu & Birhane for some examples). Even just a few misogynistic and verifiably pornographic images are harmful.

To me, the level of risk presented here seems too high for the proposed contribution (more conversational image-text models) without further mitigation steps. I would appreciate further review from ethics experts.


---
EDIT (07/16/2021): The authors have provided detailed updates and alleviated many of these issues, through clarification, bias testing, and the decision to remove detected faces - see our discussion. After our discussion, only these concerns remain:

**Remaining NSFW images.** The authors point out the immense cost of manually filtering out the estimated 700-1000 NSFW images remaining in the dataset, judging that the cost (~$50k) is not worth the additional ethical benefit. I would encourage the authors to view this as a risk that must be mitigated, not an additional benefit: NSFW images could include (and have in the past, see Prabhu & Birhane) hate speech/imagery or non-consensual pornography. The risk of amplifying - through distribution and model behavior - non-consensual pornography, derogatory speech/imagery, and offensive content is much greater than the cost of manual filtering. As an exaggerated example, consider "revenge porn" lawsuits, which can be settled for amounts [orders of magnitude greater](https://nypost.com/2018/04/10/revenge-porn-victim-gets-record-6-45m-settlement/) than $50k. Automated filtering is a great efficiency-multiplier and it would be nice if those methods improved, but there really is no margin for error on this issue - that's why a company like Facebook [employs and funds](https://www.theverge.com/2019/2/25/18229714/cognizant-facebook-content-moderator-interviews-trauma-working-conditions-arizona) ~15k content moderators. (On that note, relying on Reddit content moderation is another great efficiency multiplier, but not sufficient - to rely solely on Reddit moderation to clean up the remaining images is to defer the responsibility for the dataset's content to Reddit users and moderators.)

**Representation bias / stereotyping.** The authors did a great job with an example audit testing for gender stereotypes in the dataset, finding that much of the content is male-dominated. I expect this will be included with the dataset documentation and referenced in Section 3.2. At some point, the dataset should also be evaluated for racial/ethnic stereotypes in a similar fashion. One other way around this issue would be to remove all identified people from the dataset, since the dataset is "not intended to depict or describe people." If those kinds of images are needed (e.g. for person detection), the authors could adopt an opt-in strategy for just images containing people.

**Ethical use & documentation.** In our discussion, the authors clarified the intended use of the dataset:

>  We believe that the dataset should only be used for scientific investigation into large-scale vision+language tasks, which encompasses a wide variety of possible research use-cases. We do not believe that the dataset should be used to train models that will actually be deployed to classify or otherwise make decisions about users or user data as part of a product offered by businesses, government agencies, etc. Since the dataset has not been carefully examined for all possible representational biases, we especially warn against uses of models trained on RedCaps that classify images containing people.

I expect this will be prominent in the documentation/datasheet, especially under Q48 ("Will the dataset be distributed under a copyright or other intellectual property (IP) license, and/or under applicable terms of use (ToU)?"). I don't think this dataset should be released without a license specifically prohibiting commercial use. Ideally the authors could also prohibiting use in automated decision-making about individuals, as per the authors' own statement above, with something like a [Montreal Data License](https://arxiv.org/abs/1903.12262) or [RAIL](https://www.licenses.ai/). Enforcement of licenses like these is tricky, but the statement counts.

**Relation To Prior Work:**

In 5, the authors briefly summarize other image-text datasets, but only hint at the differences in their own work. The introduction (1) is more clear: this dataset is unique in its size, continuity, and conversationality.

Given the large portion of the work spent on ethical considerations (3.2), the authors may also consider citing work around the construction of ethical datasets, or other image datasets constructed with ethics in mind (e.g. [FairFace](https://github.com/joojs/fairface), or the revised version of ImageNet). (The authors did a good job of applying a relevant method in 3.2, but more discussion might be warranted.)

**Summary And Contributions:**

The authors present a new large-scale image-text dataset collected from subreddits devoted to images. This dataset provides a set of images and captions more actively and realistically curated than typical image datasets, without complex automated filtering. The authors propose several methods to mitigate bias concerns and show evidence that models trained on this dataset produce more attractive captions.

By way of framing, my expertise is in ethical computer vision - hence the focus on ethics in this review. I am very confident in the ethics-related critiques, less so in the critiques related to data quality for image captioning and transfer learning tasks.

---
EDIT (07/16/2021): I'm upgrading my review to 5 (marginally below acceptance) thanks to the authors' responses. The leftover margin is delineated in the Ethics section - if these concerns were fully resolved, my rating would change to >7.

---

> ### Author Response · Authors · 2021-07-14
> **Reviewer ugYV response (1)**
>
> **Privacy Risks:** Before responding to individual points, we want to emphasize one key point that threads through several of the privacy concerns raised by the reviewer: all images and captions in RedCaps are already publicly available on Reddit. Therefore RedCaps does not reveal any information about any individual that was not already public. Furthermore we do not augment RedCaps images with any user information, labels, or annotations beyond those already publicly available on Reddit, and searching for personally identifiable information in RedCaps images is no easier than searching for the same information directly via Reddit APIs.
>
> As such, the only privacy risk we see in RedCaps is to potentially give additional exposure to images or information that was already public. While we hope that our dataset will be used by many computer vision researchers, realistically this community is small compared to Reddit’s user base, since it is one of the most trafficked websites on the internet. We therefore expect that the additional exposure received by any particular image due to its inclusion in RedCaps will be marginal in comparison to that it has already received by appearing on Reddit.
>
> This reasoning colors our approach to privacy in RedCaps – we aim to minimize potential exposure of individuals’ identifiable data where possible (e.g. through subreddit selection, removing images with faces, opt-out mechanism) but since we feel the absolute privacy risk even of unfiltered RedCaps data is low, we do not attempt to completely eliminate private information when doing so would be prohibitively costly or difficult.
>
> **Usefulness of stylistic captions:** We envision that a primary use-case for RedCaps will be generic multimodal pretraining of image and textual representations that transfer to downstream tasks. From this perspective, stylistic captions give a rich semantic learning signal vs the drier text that appears in prior large-scale datasets; this is evidenced by our experiments that RedCaps-trained models outperform those trained on SBU or CC-3M for many zero-shot and low-shot recognition tasks (_Table 4_).
>
> **Documentation:** We will revise the intended uses section of the datasheet and apologize for confusion. We believe that the dataset should only be used for scientific investigation into large-scale vision+language tasks, which encompasses a wide variety of possible research use-cases. We do not believe that the dataset should be used to train models that will actually be deployed to classify or otherwise make decisions about users or user data as part of a product offered by businesses, government agencies, etc. Since the dataset has not been carefully examined for all possible representational biases, we especially warn against uses of models trained on RedCaps that classify images containing people.
>
> We believe that our image captioning experiments are consistent with these recommendations. Our image captioning models were trained for scientific investigation; no predictions from these models could not have any significant impact on the lives of users or people appearing in images (e.g. decisions about mortgages, job applications, criminal sentences; or moderation decisions about user-uploaded data that could result in bans from a website, etc). A small set of annotators judged predictions from our models, but this experiment is certainly not a production use-case.
>
> **Promised Safeguards:**
> There are two types safeguards described in _Section 3.2:_
>
> 1. Opt-out form: This was not implemented at the time of submission (since the project website was not finalized) but it is now [available here](https://forms.gle/2Us8D1FxXpEdWHi17).
> 2. Data filtering: These are already implemented - we have already run the proposed filters on the entire dataset and presented analysis of the results in _Section 3.2_. As promised we will only release the “filtered” version of the dataset. We also retrained small models on filtered data, refer [general comments to all reviewers](https://openreview.net/forum?id=VjJxBi1p9zh&noteId=W2k6pzunTe0).

---

> > ### Author Response · Authors · 2021-07-14
> > **Reviewer ugYV response (2)**
> >
> > **Training on unfiltered data:** We respectfully disagree with the characterization of the unfiltered dataset as “containing high levels of NSFW images, faces, and derogatory language.” - per _Table 3_, we estimate that **<0.01%, 0.01%**, and **2.3%** of images respectively fall into these categories; we estimated these  based on manual examination of 50K unfiltered images, in which we found only **3-4** images containing potentially NSFW or derogatory content; these are hardly "high levels”. That **2.3%** of images contain detected faces is also significantly lower than other datasets, e.g. 8-10% of ImageNet as estimated by [Prabhu and Birhane, 2020].
> >
> > All instances presented to crowd workers are [available here](https://web.eecs.umich.edu/gkaul/redcaps/human_eval/). We manually inspected these instances and found no examples of harmful stereotypes or NSFW images or language. We found 28/1000 images containing unblurred faces, and of those 13 were images of public figures (actors, politicians) or highly publicized “newsworthy events” (e.g. images of rioters at the US Capitol on 1/6/2021). Due to the public nature of the original images and the low absolute privacy risk described above, we feel that presenting a small number of images with unblurred faces to three crowd workers each has extremely low risk.
> >
> > As described in [our comment to all reviewers](https://openreview.net/forum?id=VjJxBi1p9zh&noteId=W2k6pzunTe0), preliminary experiments with pre-training on filtered data show only a small drop in downstream task performance vs training on unfiltered data.
> >
> > **Bias Testing:** We agree that this is an important topic to consider and thank the reviewer for the concrete recommendations. As suggested, we have replicated the gender-based analysis of _[Wang et al, 2020]_ on a subset of RedCaps data (including images with faces). Some of their analysis relies on ground-truth object and scene annotations which are unavailable on RedCaps, so we predict objects and scene categories using Faster R-CNN R50-FPN and ResNet-18 models trained on COCO and Places365 respectively. We follow their methodology for assigning gender to images by looking for gendered nouns in captions. [Results are shown here.](https://web.eecs.umich.edu/gkaul/redcaps/dataset_biases/)
> >
> > We emphasize that these statistics are based on detected rather than ground-truth object occurrences, and that detections are extremely noisy -- per _Table 3_, the face detector only has a precision of **~40%**, meaning that the majority of detected faces are false positives. We expect that object detections may be similarly noisy. This noise means that our computed statistics may not accurately reflect gender biases in RedCaps data. The gender co-occurrence statistics for both scenes (“gender contexts”) and object categories (“object co-occurences”) shows a male bias for nearly all scene and object categories. We hypothesize that this may be partially explained by the male bias in Reddit’s userbase (`L253`). Object interaction statistics are more gender-balanced, but some object categories (“kite”, “stop sign”, “cow”) show significant gender bias.
> >
> > We appreciate the pointer to the inclusive people annotations by _[Schumann et al, 2021]_, but their work is not directly applicable to RedCaps since they describe a method for collecting ground-truth person annotations, and we do not provide any such ground-truth object annotations in RedCaps.
> >
> > For the specific case of the beachporn subreddit, we emphasize that it is jokingly titled and is a space for users to share non-pornographic images of beautiful beaches. The subreddit FAQ specifies that the focus of the image should be the beach itself; images may contain people, but images whose focus is people should be submitted elsewhere.
> >
> > However, due to concerns about potential gender bias, we have decided to remove all images with detected faces from RedCaps. Our dataset is not intended to depict or describe people (`L90-91`), and we hope that minimizing the number of identifiable people in the dataset can help reduce the impact of unintended representational biases.
> >
> > **Deferred responsibility:** We agree that distributing images URLs together with detected face bounding boxes cannot guarantee that dataset users will blur faces. Therefore, we have decided to remove all images with detected faces from RedCaps.

---

> > > ### Author Response · Authors · 2021-07-14
> > > **Reviewer ugYV response (3)**
> > >
> > > **Opt-out vs opt-in:** Any dataset comprising previously private data should obviously be strictly opt-in. However due to the already public nature of all data in RedCaps, we feel that our opt-out policy is sufficient. In an ideal world opt-in may be preferable to opt-out, but in practice we are unaware of any large-scale web image dataset that has adopted policies of user notification or opt-in. We believe that our explicit opt-out policy is already an improvement over existing datasets which are typically neither opt-in nor opt-out. This includes the two “image datasets constructed with ethics in mind” mentioned by the reviewer: FairFace _[Karkainen and Joo, WACV 2021]_ samples images from YFCC-100M, and revised versions of ImageNet _[Yang et al, FAT* 2020, Yang et al, arXiv 2021]_ remove or modify images from the original ImageNet dataset; neither dataset notified users or adopted either opt-in or opt-out policies.

---

### Official Review · Reviewer_HHUr · 2021-07-04
**A large scale vision-language corpus collected from subreddits.**

**Rating:** 7
**Confidence:** 4
**Correctness:** The paper is technically sound.
**Clarity:** The paper is well written

**Strengths:**

1. The dataset collection is well described and motivated. Reddit has traditionally been a source of high-quality language content and is relatively well moderated.

2. The authors clearly mention the dataset collection procedure. The subreddit selection ensures that the content does not contain memes, sketches, and, most importantly, too many celebrity pictures. The caption sanitization step also ensures that the text is processed to remove unwanted characters, emojis, non-Latin characters, insta-id, self-promoting text, etc.

3. The dataset statistics are adequately reported.

4. The ethical concerns are discussed reasonably well. The authors have removed children's faces using automatic age detection methods (some examples will probably remain) and have also removed harmful text content using several techniques. They also will have a protocol to remove content uploaded without consent manually once brought to their notice.

5. The experiments conducted help establish that the dataset can be used and meaningful signals can be learned from it for several tasks.

6. This will be one of the largest publicly available corpus for vision-language and can benefit the community.

**Weaknesses:**

1. While pretraining on RedCaps is hugely beneficial for Zero-shot classification, and low shot classification, the scores in other tasks are very similar to CC-3M while CC-3M having only 1/4th data. Is there any specific reason for this? Does CC-3M has data that is better suited for these tasks?

2. Qualitative results for some tasks like Instance segmentation are missing from both the main paper and the supplementary.

**Additional Feedback:**

Currently, the dataset does not have any comparison with CC-12M. I believe CC-12M was released just before the submission, and it might have been impossible to add comparisons with it during the first-round submission. Since CC-12M has more comparable size to RedCaps, adding comparison results for it will in the main paper or supplementary materials will be insightful.

**Documentation:**

The dataset collection procedure is clearly mentioned and additional information is provided in the supplementary materials

**Ethics:**

The authors have described them well. As they pointed out, such a large corpus collected from the internet will always have some biases but I believe the positives of the data outweigh the risks.

**Relation To Prior Work:**

Prior work is mentioned adequately

**Summary And Contributions:**

The authors propose a dataset called RedCaps which consists of 11.7 Million image-text pairs collected from 256 subreddits. The data includes popular day-to-day items in social media that are shared by Reddit users regularly. The authors describe their data collection pipeline, which includes selecting subreddits based on the volume of image posts, filtering out posts based on upvotes, and cleaning captions for better image-text mapping. In the paper, the authors have reported different corpus statistics and have used the dataset for different vision-language tasks. The evaluations show that the RedCaps dataset is useful for pre-training feature extractors that can be used for multiple downstream tasks like image captioning, zero-shot image classification, etc.

---

> ### Author Response · Authors · 2021-07-14
> **Reviewer HHUr response**
>
> **Pre-training is beneficial for zero-shot and low-shot but not for other tasks?**
> Excellent question! The benefits of pre-training shine the most when additional training data for downstream tasks is very less (low-shot transfer), or not available at all (zero-shot transfer) – we can observe this in _Table 4_, where RedCaps pre-trained models have very large improvements over SBU and CC-3M.
>
> On the other hand, the benefit of pre-training is less visible when the downstream task has adequate training data. We observe this in _Table 5_ with tasks that use large datasets – ImageNet classification (**1.28M** images) and COCO/LVIS segmentation (**118K** images). Prior works in self-supervised pre-training have also observed this trend – for example, MoCo [(He et al, CVPR 2020)](https://arxiv.org/abs/1911.05722) observed _less than 1 AP_ improvements on COCO/LVIS segmentation despite pre-training on 1B images from Instagram – **1000x** more pre-training data than ImageNet. We will add this discussion in our paper.
>
> **Qualitative examples:** Thank you for your suggestion, we will add them in the supplementary in our camera ready version!
>
> **[Additional Feedback] Comparison with CC-12M:** You are correct, CC-12M was released very close to the NeurIPS deadline, it was computationally impossible for us to train models with it. This is why we included a model trained on RedCaps-2020 subset, which is of similar size as CC-3M (`L265-266`). We will add comparisons with CC-12M in our camera-ready version.

---

### Official Review · Reviewer_N5hZ · 2021-07-05
**Review for RedCaps**

**Rating:** 8
**Confidence:** 4
**Clarity:** Yes. The paper is well written.

**Strengths:**

+ RedCaps is built upon Reddits where the image-text pair is generated by users and thus potentially simplifies data collection procedure and increases data quality.

+ The paper conducts extensive experiments on visual-textual representation learning. In particular, It applies the learned representation to ten different downstream visual recognition tasks.

+ The data collection procedure is clearly described. The pipeline including image post filtering and caption sanitization is discussed in details.

+ The paper discusses ethical concerns, including user contents, privacy (face/children), and harmful stereotypes.  It also provides solutions to alleviate such ethical concerns.

**Weaknesses:**

- In order to align with user content,  posts deleted from Reddit will be removed from RedCaps. This may be problematic when comparing a new method with a previous one (since data used to training may be less).


====after rebuttal====

The authors have well addressed my concerns. It is a good idea to only include the posts that exist at least 6 months in the dataset since those are unlikely to be deleted.

**Additional Feedback:**

No.

**Correctness:**

The data collection procedure is clearly described. The pipeline including image post filtering and caption sanitization is discussed in details.

**Documentation:**

Yes. The paper provides sufficient details on data collections, availability and maintenance, and ethical issues. It also attaches a project page illustrating data.

**Ethics:**

The paper discusses ethical concerns, including user contents, privacy (face/children), and harmful stereotypes.  It also provides solutions to alleviate such ethical concerns.

**Relation To Prior Work:**

Yes, the paper clearly discusses the relation to prior works.

**Summary And Contributions:**

The paper introduces a large-scale image-text pair datasets called RedCaps, sourced from Reddits. The paper argues for advantages of Reddits as a source of data compared with previous ones such as search engines. Based on the image-text dataset, the authors conducts experiments and show that the learned representation is transferrable to many downstream tasks. The dataset can also be used for image captioning .

---

> ### Author Response · Authors · 2021-07-14
> **Reviewer N5hZ response**
>
> **Image removal from RedCaps makes comparison between old and new models unfair:**
> We agree, but note that it is a necessary trade-off to respect user privacy (`L220–222`). Based on your concern, we tried to estimate the amount of image removal in RedCaps.
> We checked all image URLs for RedCaps-2020 subset (**2.84M** instances). We found that **32K** images (**1.09%**) were removed as of July 9, 2021. Looking closely within three-month durations:
>
> | Duration (in 2020) | Size at time of collection (February 2021) | Images disappeared as of July 9, 2021 |
> |:-------------------|:----:|:----:|
> | January – March    | 660K | 5639 (0.85%) |
> | April – June       | 819K | 7943 (0.96%) |
> | July – September   | 737K | 8586 (1.16%) |
> | October – December | 626K | 9098 (1.45%) |
>
> We find that most removals are from Oct-Dec 2020. These posts were only 2-4 months old when we collected them in Feb 2021. The trends show that the likelihood of image post deletion reduces as more time goes by (> 6 months), hence we expect removals from 2019 and earlier to be fewer than these. Following these observations, we will collect data for future versions of RedCaps at least 6 months after the end of year (e.g. we will collect and release RedCaps-2021 in July 2022).
>
> As for the dataset release upon acceptance, we will remove these instances while retraining models with face-blurred images, and recompute data statistics for our analysis. Based on our retraining experiments with the **~2.3%** smaller RedCaps `NO-FACES` subset (Refer [overall comments](https://openreview.net/forum?id=VjJxBi1p9zh&noteId=W2k6pzunTe0)), we think that this **~1%** size reduction will not significantly affect performance of newly trained models. Note that all public datasets similar to RedCaps (_Figure 3, bottom_) also distribute images as URLs. While they are _static_, RedCaps will continue to grow over time as new content gets uploaded on Reddit – the trends in (_Figure 3, top_) indicate that RedCaps will have **more than 14M** instances by the end of 2021.

---

### Author Response · Authors · 2021-07-14
**Thank you all reviewers! We made an update to RedCaps: removed all detected faces.**

We thank all the reviewers for their thoughtful and detailed feedback! We are glad that all reviewers liked multiple aspects of our paper including simple and reproducible data collection, extensive experiments, adequate discussion on alleviating ethical issues, and the clarity in our writing/presentation. We first discuss an update in RedCaps that further minimizes ethical risks, and respond to reviewer concerns individually.

**Update: Removing Images with Detected Faces.**
Due to concerns about gender representation and deferred responsibility raised by Reviewer `ugYV`, we have decided to remove all images with detected faces from RedCaps rather than implore users to blur them.
Per _Table 3_, this will remove **~272K** images from RedCaps, reducing the dataset size by **~2.3%**.

We expect that this will not significantly affect downstream task performance for visual representations trained on RedCaps data. Training full models takes ~13 days and was not possible during the rebuttal period, but we have validated this idea by training smaller models (L=3 Transformer layers vs L=6) for fewer iterations (600k vs 1.2M). We trained models on three different versions of RedCaps-2020 data:

1. `UNFILTERED`: Unfiltered data, as was used for training in the submission.
2. `FACE-BLURRED`: Uses the filtering steps described in _Section 3.2_: Instances with NSFW images, harmful language, or child faces are removed; adult faces are blurred.
3. `NO-FACES`: Uses the same filtering steps as (2), but removes images with adult faces rather than blurring them.

Downstream performance on some tasks is reported below. We selected these tasks based on fast evaluation speed.
For VOC classification, we report overall mAP and "Person" class AP. For Pets, Food and Flowers, we report zero-shot Top-5 accuracy.

| SPLIT | VOC mAP | VOC Person AP | Pets | Food | Flowers |
|:--------------|:----:|:----:|:----:|:----:|:----:|
| `UNFILTERED`  | 84.7 | 96.5 | 54.7 | 63.7 | 42.7 |
| `FACE-BLURRED`| 84.1 | 96.3 | 51.2 | 70.1 | 42.9 |
| `NO-FACES`    | 83.5 | 96.0 | 53.6 | 66.5 | 42.7 |

We observe that removing all faces does not significantly degrade performance on these tasks. For example, removing faces degrades VOC Person AP by **0.5** – this indicates that visual features pre-trained on RedCaps images without faces can still be used to train linear models (or fine-tuned) to detect presence of people.
Moreover, `NO-FACES` performs better on some tasks that do not involve people (e.g. Food).

We will use the extra space in the camera-ready version of the paper to expand upon these results with full training runs and results on all downstream tasks.

---

### Comment · Reviewer_Abkv · 2021-08-06
**An official ethics comment**

This is a comment from one of the official ethics reviewers.

The authors should be commended for their original submission's focus on the issues raised by Prabhu and Birhane. Reviewer ugYV should be equally commended for being so fastidious in their review and back-and-forth with the authors, who were thoughtful and responsive throughout. All the actions that the authors have taken (not just promised), including removing the 2.3% of images with detected people, have continued to strengthen the ethical aspects of the work.

The remaining issue is the calculus of the NSFW filtering cost ($50K) and ethical benefit. Note that this will not be a one-time cost, but an ongoing one as long as the dataset is maintained. Although we may not like to assign monetary amounts to ethical benefits, there is no way around it. I do sympathize with the thought that we should have exactly zero NSFW examples in the dataset, but I also believe that nothing in real life is ever perfect. I am satisfied by the current efforts of the authors and given the strong technical reviews, am not opposed to this work being published.

---

### Comment · Reviewer_Jurh · 2021-08-12
**An Official Ethics Review Comment**

This work includes user-generated content that can include privacy and consent issues, including for vulnerable populations (faces, and children).  The concerns mentioned also include NSFW images and harmful stereotypes.

The authors do a fantastic job at highlighting some of the key ethical concerns in this work (consent, privacy, stereotypes, demographics) and do due diligence on addressing them.  Another ethical concern that the authors speak to, but don't name directly, has to do with "proliferation" and amplification of images that were not intended to be shared outside of reddit.  (This concern is implicit in the other listed concerns).  The authors address this by utilizing urls, rather than the images themselves, which allows reddit users to delete their content and have that content correspondingly deleted in the dataset.

The authors even provide a datasheet!  And additional metadata about the dataset beyond that.  Would be great to make sure these are easily accessible at the host url (https://vhosts.eecs.umich.edu/kdexd/redcaps/).

Regarding NSFW images, I wouldn't agree that these bring up serious ethical concerns as long as the other related concerns around consent and privacy are reasonably addressed, as they seem to be.

Regarding demographic skews and stereotypes, as long as these issues are made clear, they are reasonable aspects of this dataset. If for no other reason, these provide insight into the skews within Reddit, and are useful for describing what Reddit is like.  In my opinion, it is the work of people who use this dataset to train models, rather than the dataset creators, to address skews and stereotypes according to their models' intended use.

Since the initial reviews, the authors have done even more work on addressing the ethically problematic issues.

I think this dataset is reasonable to release, as long as the other reviews agree the work as a whole is sound.

---

### Comment · Reviewer_4G3Q · 2021-08-13
**Dataset with much more due diligence compared to prior computer vision datasets**

I commend the authors for thinking through the ethical considerations, making their various choices explicit and documenting them in a data sheet. While there are still concerns, in particular the potential presence of stereotypes and derogatory images, the presence of children's faces, I have not seen this level of diligence in other large scale datasets and see this due diligence as a positive step forward. I think reviewer ugYV has very thoroughly listed out the concerns and their gravity, and I agree with all of their analyses. Unfortunately, I believe the computer vision community is extremely behind in understanding this, and compared to prior datasets in the community, this one does a much better job of thinking through the concerns, attempting to mitigate them and making explicit their choices.

I have some additional comments below

“We only collect 100 posts with images from three image hosting domains: Reddit (i.redd.it), Imgur (i.imgur.com), and Flickr (staticflickr.com). "

Be aware that IBM, Microsoft etc are in a lawsuit for violating Illinois biometric laws by creating the diverse faces dataset using Flickr and applying it to applications of automatic facial analysis. Even though people uploaded their photos to Flickr they did not consent to having it used in large scale automated facial analysis tools. As mentioned by Reviewer ugYV below, even though the cost of curation can seem high, the cost/potential harm of not curating, and even 1 child's face being present in the dataset can be really high. While all the steps the authors have taken are much more than prior computer vision datasets, and probably current datasets that may be published in computer vision venues, I suggest that the authors do as much as they can to ensure that there are zero, rather than potentially a few, children's faces (e.g.)

“Subreddit sizes highly correlate with their 151 popularity on Reddit, which depends on what humans find interesting to view and share on social media.”

I suggest that the authors change this to qualify to Reddit users rather than all humans. They mention in their paper that size doesn't guarantee diversity. Similarly, Reddit users' interests can't be generalized to "what humans find interesting."

I suggest some way of limiting the dataset usage by either having a form where people request to use the dataset for specific purposes, a lisence etc.

---

### Note · ~Karan_Desai1 · 2021-11-07

https://redcaps.xyz

---

### Decision · Program_Chairs · 2021-07-26

Accept